# Epidemiology of Hepatitis C virus infection among incarcerated populations in North Dakota

Liton Chandra Deb[1,2], Hannah Hove[1,3], Tracy K. Miller[4], Kodi Pinks[4], Grace Njau[4], John J. Hagan[5], Rick J. Jansen[1,6,7,8]*

1 Department of Public Health, North Dakota State University, Fargo, ND, United States of America, 2 Department of Population Health and Pathobiology, College of Veterinary Medicine, North Carolina State University, Raleigh, NC, United States of America, 3 University of North Dakota School of Medicine & Health Sciences, Grand Forks, ND, United States of America, 4 North Dakota Department of Health, Bismarck, ND, United States of America, 5 North Dakota Department of Corrections and Rehabilitation, Bismarck, ND, United States of America, 6 Genomics, Phenomics, and Bioinformatics Program, North Dakota State University, Fargo, ND, United States of America, 7 Center for Immunization Research and Education (CIRE), North Dakota State University, Fargo, ND, United States of America, 8 Center for Diagnostic and Therapeutic Strategies in Pancreatic Cancer, North Dakota State University, Fargo, ND, United States of America

* rick.jansen@ndsu.edu

**Data Availability Statement:** Data cannot be shared publicly because of the sensitive nature of justice involved individual data as determined by the North Dakota Department of Corrections

## Abstract

This retrospective cohort study was conducted to determine the prevalence of HCV infections among individuals incarcerated in a state prison system and identify potential contributing factors to HCV infection. North Dakota Department of Corrections and Rehabilitation (NDDOCR) data from 2009 to 2018 was used and period prevalence was calculated for this 10-year time period. The period prevalence of HCV infection was (15.13% (95% CI 14.39–15.90) with a marginally significant (p-value: 0.0542) increasing linear trend in annual prevalence over this period. Multivariate logistic regression analysis was used to identify risk factors associated with HCV infection. The main significant independent risk factors for HCV infection in this incarcerated population were age >40 years [OR: 1.78 (1.37–2.32)]; sex [OR: 1.21 (1.03–1.43)]; race/ethnicity [OR: 1.97 (1.69–2.29)]; history of intravenous drug use (IVDU) [OR: 7.36 (6.41–8.44)]; history of needle or syringe sharing [OR: 7.57 (6.62–8.67)]; and alcohol use [OR: 0.87 (0.77–0.99)]. Study limitations include uncollected information on sexual history, frequency or duration of injection drug use and blood transfusion history of the incarcerated population. Considering the high prevalence of HCV infection and its associated risk factors, it is important to implement prevention programs such as syringe/needle exchanges and counsel with imprisoned IVD users.

## Introduction

Hepatitis C virus (HCV) infection is the most frequently reported blood borne infection in the United States in both general and incarcerated populations [1]. HCV causes both acute and chronic infection. Acute HCV infection mainly occurs within the first 6 months after a person

Research Ethics Committee. However, data can be obtained for researchers who meet the criteria for access to this confidential data (contact via Kayli Richards karichards@nd.gov).

**Funding:** Funding for this project provided by the Department of Health and Human Services (https://www.hhs.gov/, G17.1102, TM) with subaward issued to RJJ. The funders had no role in study design, data collection and analysis, decision to publish, or preparation of the manuscript.

**Competing interests:** The authors have declared that no competing interests exist.

is exposed. Although acute infection can be a short illness, in the majority cases it leads to a lifelong or chronic infection. HCV infection is also considered a major disease worldwide with a prevalence of almost 3%, which means that more than 170 million people suffer with chronic hepatitis C [2]. It is considered one of the world's current significant health problems as HCV leads to severe liver complications and early mortality [3].

Based on a US study between 2003 and 2013 examining national multiple-cause-of-death (MCOD) data, the annual HCV related deaths were increasing since 2007 with an estimate that 19,368 people died due to HCV in 2013[4]. From 2003 to 2013 the number of deaths associated HCV surpassed the total number of deaths from 60 nationally notifiable infectious diseases combined [4]. In fact, in the US, a national health survey of data collected from 2003 to 2010 indicated that approximately 3.5 million people were infected with HCV with an overall incidence rate 1.0 cases per 100,000 population [5].

Previous studies have indicated that incarcerated populations have a disproportionately higher risk to HCV exposure and infection compared to the general US population. The existing studies on incarcerated populations have indicated that HCV antibody positivity rates range from 12 to 34%, which is over 20 times the national level [6–10]. As previously mentioned, HCV infection disproportionately affects those who have been in jail and prison [11]. Unfortunately, the data for chronic HCV infection in incarcerated populations is scarce and not current in the US. Still existing data indicates a high rate of infection and disease burden among the incarcerated population. For example, in 2002, Hammett et al. indicated that US correctional populations in 1997 accounted for 29.4% to 43.2% of total HCV infections [12]. Also, the CDC database for 2003 derived from HCV surveillance data from eight states reported that 16% to 41% of prison populations in those eight states had serological evidence of prior HCV exposure and the HCV cases among correctional population was responsible for 1.3 million (39%) of the HCV burden within those states[13]. Thus, based on published reports, HCV infection prevalence is much higher in prison populations than in general populations because of a much higher prevalence of risk-factors among the prison population [14].

Risk factors of HCV that are observed at a higher rate in incarcerated populations compared to the general population includes heavy alcohol drinking, IVDU, and needle sharing. Heavy alcohol drinking has been reported to increase the risk of HCV by 1.58 times [15], IVDU is reported in a meta-analysis study to increase the risk of HCV infection by 24 times [16], while needle sharing has been associated with an increased risk of 2.58–4.06 times [17, 18].

There is a lack of current data on HCV infections and risk factors in incarcerated people in the US. Therefore, this study was conducted from 2009 to 2018 in North Dakota Department of Corrections and Rehabilitation (NDDOCR) to update data on the prevalence of HCV infections in incarcerated people in the US and identify possible contributing risk factors.

## Methodology

### Study population

This is a retrospective cohort study of 8,836 incarcerated people in NDDOCR from 2009 to 2018. Of the total 8,836 justice involved individuals, 1,337 were chronic HCV virus and 7,499 were HCV virus negative when testing for antibodies to HCV (anti-HCV) in serum. Reflex testing with universal opt-out was completed upon arrival for HCV antibody and if positive, viral load and genotype identification was performed. All 8,836 incarcerated persons were unique individuals; repeated persons were entered in the database as a single incarcerated person and only counted once in this database. The study protocol, and materials were reviewed and approved by the ND Department of Health Institutional Review Board. As in the original study (Substance Use and early Death Among Justice Involved Individuals of the ND Dept of

Corrections and Rehabilitation, 2010–2018 (ND-045-052019)), informed consent was not collected, due to minimal risk associated with this study. Data used for this study were the de-identified data set created from the original study.

## Data collection

Demographic, medical, behavioural, and incarceration-related information was collected from NDDOCR with the help of North Dakota Department of Health (NDDoH). Medical information consisted of the history of intravenous drug use, abnormal liver test results, and screening results of HCV (testing for antibodies to HCV within asymptomatic incarcerated persons). Behavioural information included the history of alcohol consumption, drug use, and needle sharing. The needle sharing variable was assessed using the question, "Now or in the past, have you shared injection drugs or needles with anyone else? This includes needles, syringes, spoons, filters or rinse water?" The intention was to identify opportunities to transmit HCV during the injection process. Demographic information included birth date, sex, and race/ethnicity. Also, incarceration-related variables included length of total lifetime incarceration, number of times incarcerated, and the age at first incarceration. All behavioural, medical and incarceration related information was routinely collected for each individual at the time of their entry into the correctional facilities. In case of repeated offenders their first occurrence was only included in the study sample.

## Statistical analysis

Collected demographic, medical, behavioural, and incarceration-related information were initially entered into Microsoft excel spreadsheet and coded for analysis. Statistical Analysis System (SAS) version 9.4 (SAS Institute Inc., Cary, NC, USA) and R statistical program version 4.1.2 were used to perform statistical analyses and generate tables and figures. Age was categorized into 3 groups: less than 20, 21 to 40 and more than 40 years old. Age was categorized into three groups based on the previous literature detailing that people born during 1945–1965 (i.e., baby boomers) are at higher risk of HCV as a group because of potential contaminated blood exposures. Race/ethnicity was coded into 5 groups: Black (black and African American), Caucasian, Native American, Hispanic and others. All the other variables were categorized based on yes or no response. Prevalence with 95% Confidence Intervals (CIs) were calculated on the basis of binomial distributions. Summary statistics included frequency tables, for categorical variables, medians and 95% CIs. Associations between HCV antibody status and demographic, risk exposure–related, and incarceration-related variables were conducted using the $\chi^2$ test of association (or Fisher's exact test, in cases for which the expected cell size was less than 5 events), and Odds ratio (ORs) and 95% CIs were calculated. Independent associations between the various exposures and HCV antibody were assessed using unconditional logistic regression analyses. Explanatory variables with p-values $\leq 0.10$ on bivariate analysis, were included into a backward stepwise logistic regression analysis. Select interactions were also checked based on previously published observations and our study results. Confounding effect of an explanatory variable was also evaluated by assigning the change of parameter estimates before and after removal of a variable from the model. If the parameter estimates of a variable increased or decreased $\geq 10\%$ after removing another variable from the model, then this one explanatory variable was considered to have a confounding effect on the outcome variable. Regression coefficients were converted into odds ratios (ORs; $e\beta$) and their 95% confidence intervals (CIs). All tests were considered statistically significant at (P $\leq 0.05$) unless otherwise noted above.

## Results

Descriptive analysis of all variables was completed and presented in Table 1. Here, 1,337 individuals were diagnosed with HCV out of 8,836 who were housed by the NDDOCR during the study period, for a period prevalence of 15.13% (95% CI 14.39–15.90). HCV diagnosis was based on incoming blood tests and antibody test (genotype confirmatory tests) with acute versus chronic determined by persistence on 6 month recheck (approximately 100% chronic). Of the 8,836 incarcerated people, 1,553 individuals were excluded from the risk factors analysis due to lack of behavioral information such as their history of alcohol use, needle sharing and drug use. Finally, 7,283 individuals were used for further analysis: 1,256 individuals were HCV antibody (Ab) positive and 6,027 were HCV antibody (Ab) negative on screening. The annual HCV prevalence for the years 2009–2018 are presented in Table 1 and visualized in Fig 1. A marginally significant (p-value = 0.0542) increasing linear trend was observed over the study period (Fig 1). The prevalence of HCV was significantly higher in the older age category (21.03%) compared with younger age category (13.01%) (Table 1). Also, the prevalence was much higher in people who had a history of taking intravenous drugs (36.11%) compared with those without history of intravenous drugs (6.98%). Similarly, the prevalence was much higher for people with sharing needles (46.82%) compared with those that did not have any history of needle sharing (10.41%) (Table 1).

Information on 7,283 incarcerated individuals was used in this analysis with 1,256 HCV antibody (Ab) positive individuals and 6,027 HCV antibody (Ab) negative individuals. Univariable analysis revealed that age, sex, race/ethnicity, birth category, alcohol use history, drug use history, history of taking intravenous drug and shared needle history were significantly associated with HCV active infection (Table 1). Hence, these eight variables were included in the multivariable analysis. In the final multivariable logistic regression model with backward elimination six variables namely age, sex, birth category, alcohol use history, history of taking intravenous drug and shared needle history are retained as significant and independent predictors of HCV status at a p<0.05. Results of the final multivariable logistic regression model were presented in (Table 2). Odds of HCV active infection was 1.78 (1.37–2.32) times higher in individuals >40 years old than in individuals ≤20 years old, 2.65 (0.6–11.65) times higher in individuals who born between 1945 to 1965, and 1.21 (1.03–1.43) times higher in incarcerated females than in incarcerated males. The odds of HCV active infection were 7.36 (6.41–8.44) times higher in incarcerated persons who had a history of taking intravenous drugs compared with non-intravenous drug users, and was 7.57 (6.62–8.67) times higher in those who had a history of shared drug injection or needle sharing. In this study, incarcerated persons with alcohol use history showed significantly lower odds of HCV active infection than those that did not have any alcohol use history after adjusting for the other five variables in the model (Table 2). Fig 1 in the supplement shows the percentage of cases which identified having a selected risk behaviour. It is observed that a majority cases had a history of taking intravenous drugs and needle sharing (S1 Fig). Potential interaction between IVDU and either sex or race/ethnicity was investigated. Although the distribution of IVDU across sex and race/ethnicity groups was different than expected using a chi-squared test (S1 Table), an interaction term in a logistic regression model (e.g., HCV ~ sex + IVDU + sex: IVDU) was not significant for either sex by IVDU or race/ethnicity by IVDU (not shown).

## Discussion

The results of this study indicate that over a period of 10 years the prevalence of HCV among incarcerated populations in ND correctional facilities was 15.13%. As expected, the rate of HCV virus seropositivity among ND incarcerated populations from 2009 to 2018 is much higher than the seroprevalence of 1.7% (95% CI 1.4–2.0%) estimated by using National Health

**Table 1. Univariable analysis (chi-square test) of plausible determinants of HCV antibody (Ab) in Incarcerated population.**

| Variables | HCV antibody (Ab) Positive n (%) | HCV antibody (Ab) Negative n (%) | HCV Period Prevalence Over Study (%) | P*-value |
|---|---|---|---|---|
| Age | | | | < .0001 |
| ≤20 year | 83 (6.61) | 555 (9.21) | 13.01 | |
| 21 to 40 years | 882 (70.22) | 4379 (72.66) | 16.76 | |
| >40 year | 291 (23.17) | 1093 (18.14) | 21.03 | |
| Sex | | | | < .0001 |
| Female | 281 (22.37) | 937 (15.55) | 23.07 | |
| Male | 975 (77.63) | 5090 (84.45) | 16.08 | |
| Race/ethnicity | | | | < .0001 |
| Black | 23 (1.83) | 399 (6.62) | 5.45 | |
| Caucasian | 641 (51.04) | 2537 (42.09) | 20.17 | |
| HIS | 45 (3.58) | 229 (3.80) | 16.42 | |
| NAT | 360 (28.66) | 724 (12.01) | 33.21 | |
| Others | 187 (14.89) | 2138 (35.47) | 8.04 | |
| Birth Category | | | | < .0001 |
| Before 1945 | 2 (0.16) | 16 (0.27) | 11.11 | |
| 1945 to 1965 | 179 (14.25) | 540 (8.96) | 24.90 | |
| After 1965 | 1075 (85.59) | 5471 (90.77) | 16.42 | |
| Time spent in incarceration | | | | 0.4749 |
| Less than 1 year | 173 (13.77) | 785 (13.02) | 18.06 | |
| More than 1 year | 1083 (86.23) | 5242 (86.98) | 17.12 | |
| Alcohol | | | | 0.0025 |
| Yes | 551 (43.87) | 2926 (48.55) | 15.85 | |
| No | 705 (56.13) | 3101 (51.45) | 18.52 | |
| Drug Use | | | | < .0001 |
| Yes | 1109 (88.30) | 4608 (76.46) | 19.40 | |
| No | 147 (11.70) | 1419 (23.54) | 9.39 | |
| History of Intravenous Drug Use (IVDU) | | | | < .0001 |
| Yes | 927 (73.81) | 1640 (27.21) | 36.11 | |
| No | 329 (26.19) | 4387 (72.79) | 6.98 | |
| History of Needle Sharing | | | | < .0001 |
| Yes | 640 (50.96) | 727 (12.06) | 46.82 | |
| No | 616 (49.04) | 5300 (87.94) | 10.41 | |
| Year of Incarceration | | | HCV Annual Prevalence (%) | < .0001 |
| 2009 | 119 (10.40) | 425 (7.69) | 21.88 | |
| 2010 | 100 (8.74) | 539 (9.75) | 15.65 | |
| 2011 | 79 (6.91) | 467 (8.45) | 14.47 | |
| 2012 | 83 (7.26) | 580 (10.49) | 12.52 | |
| 2013 | 90 (7.87) | 595 (10.77) | 13.14 | |
| 2014 | 133 (11.63) | 679 (12.29) | 16.38 | |
| 2015 | 170 (14.86) | 662 (11.98) | 20.43 | |
| 2016 | 169 (14.77) | 690 (12.48) | 19.67 | |
| 2017 | 137 (11.98) | 600 (10.86) | 18.59 | |
| 2018 | 64 (5.25) | 290 (5.25) | 18.08 | |

n, Number of people; %, percentage; HCV, Hepatitis C Virus; HIS, Hispanic; NAT, Native American

*p values were calculated using the Chi-squared test

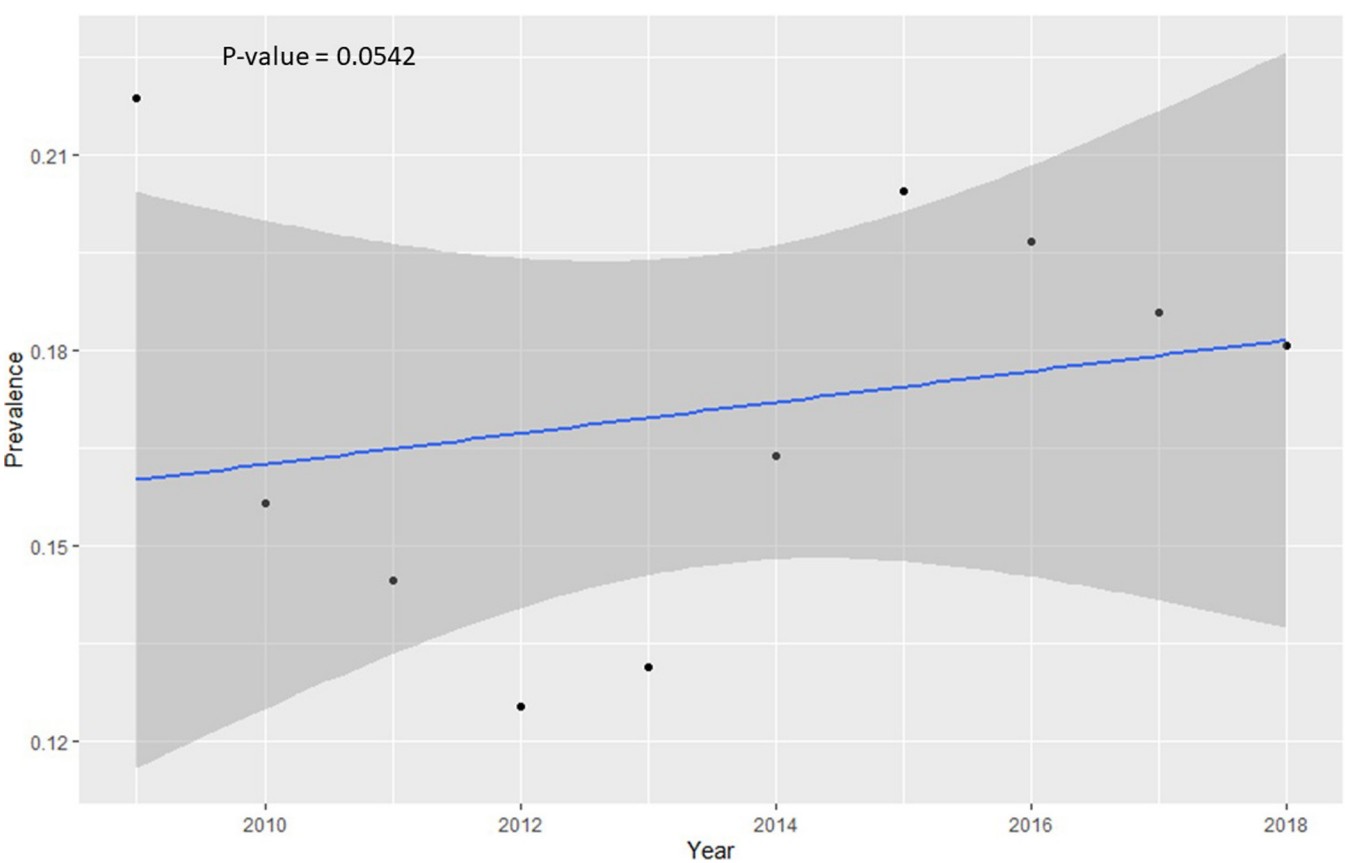

**Fig 1. Linear trend in annual prevalence across the study period 2009–2018.** The blue line represents the line of best fit and the surrounding dark grey area represents the 95% confidence band.

and Nutrition examination Survey (NHANES) for the non-institutional US population from 2013 to 2016 [19]. HCV prevalence estimates among incarcerated populations reported from the other studies such as California (34.3%) and Italy (10.4%) were also higher [6, 20]. Overall, available data suggests that HCV is highly prevalent among individuals incarcerated in prison settings [17, 21, 22]. This variation in the prevalence among the studies carried out in different regions is most likely attributed to the type of incarcerated populations surveyed on the basis of associated risk factors such as IVDU, history of imprisonment, high-risk sexual behaviours, or other high-risk behaviours. Also, as previously mentioned, HCV infection among the incarcerated population is largely attributed to their history of using IVDU. Thus, the prevalence of self-reported IVDU was higher (43%) among incarcerated populations in California as compared with the ND incarcerated population self-reported prevalence (36%) and may account for some of the difference between the HCV rates between these two states [6].

The findings of this study showed that the odds of HCV infection were 1.78 times higher among incarcerated individuals who were >40 years old compared with ≤20 years old. This finding is in line with other published investigations, including the California study that reported age independently correlated with HCV infection and that the prevalence increased with age [6]. Also, In terms of birth cohorts, the results of this study indicated that the odds of being HCV positive are 2.65 times higher in incarcerated individuals who were born between 1945 to 1965 than those born before 1945. Indeed, results from another study among US veterans are consistent this study's findings [23].

**Table 2. Final model of multivariable logistic regression analysis of plausible determinants of HCV antibody (Ab) in incarcerated populations.**

| Variables | Odds ratio (95% Confidence interval) | P-value* |
|---|---|---|
| Age | | <0.0001 |
| ≤20 year | 1 | |
| 21 to 40 years | 1.38 (1.08–1.76) | |
| >40 year | 1.78 (1.37–2.32) | |
| Sex | | 0.02 |
| Female | 1.21 (1.03–1.43) | |
| Male | 1 | |
| Race/ethnicity | | <0.0001 |
| Caucasian | 1 | |
| Black | 0.23 (0.15–0.35) | |
| HIS | 0.78 (0.56–1.08) | |
| NAT | 1.97 (1.69–2.29) | |
| Others | 0.38 (0.32–0.45) | |
| Birth category | | 0.002 |
| Before 1945 | 1 | |
| 1945 to 1965 | 2.65 (0.6–11.65) | |
| After 1965 | 1.57 (0.36–6.85) | |
| Alcohol | | 0.002 |
| Yes | 0.87 (0.77–0.99) | |
| No | 1 | |
| History of Intravenous Drug Use (IVDU) | | <0.0001 |
| Yes | 7.36 (6.41–8.44) | |
| No | 1 | |
| History of Needle Sharing | | <0.0001 |
| Yes | 7.57 (6.62–8.67) | |
| No | 1 | |

HCV, Hepatitis C Virus; HIS, Hispanic; NAT, Native American

*p values were calculated using the Chi-squared test

The results of this current ND study showed that the risk of HCV infection was higher in incarcerated females than in incarcerated males. In this study, the odds of HCV infection among incarcerated females were 1.21 times higher than for incarcerated males. Other studies have also observed that HCV infection was observed more frequently in incarcerated females than in incarcerated males [23, 24], but other studies observed higher rates among males compared to females [25]. A probable explanation behind this higher rate in females was proposed in a California study, in which a higher prevalence of HCV infected women who reported that their sexual partners were injection drug users [6]. Also, it was worthy to pointing out that those female populations' offenses more frequently related to drugs and prostitution. Unfortunately, due to the lack of data availability for ND, the sexual histories of either incarcerated males or females were not included.

We observed a significant association of HCV infection with the history of alcohol use by the incarcerated populations. Incarcerated persons with alcohol use history showed significantly lower odds of HCV infection than those that did not have any alcohol use history. This is in contrast to the National Institute on Alcohol Abuse and Alcoholism (NIAAA) report summarizing that heavy alcohol drinking increases the risk of HCV infection [26]. It is likely that our observed negative correlation was due to unmeasured confounding effects.

The results of this current study revealed that the risk of HCV infection was higher in incarcerated Native Americans (NAT). In this study, the odds of HCV infection among incarcerated NAT were 1.97 times higher than for incarcerated Caucasian. When looking at rates of IVDU for NAT and Caucasians, we also see a lower percentage of Caucasian IVDU (42.5) compared to NAT IVDU (49.1%) (S1 Table) which may partially explain the increased risk of HCV observed among NAT. Other studies observed that HCV infection was observed more frequently in incarcerated blacks [27, 28], however, we did not observe this in our study. This variation could be due to the self-reporting of race/ethnicity and the difference in minority populations across the US.

The findings of this study also showed that 73.80% of the HCV positive incarcerated populations in ND had a history of taking intravenous drugs. This finding is in line with other published investigations, including a meta-analysis study that found that male incarcerated individuals using injected drugs were 24 times more likely to be HCV positive compared with using non-injecting drugs [16]. The results from this ND study revealed that the odds of HCV infection among incarcerated populations with histories of intravenous drug use were 7.36 times higher compared with non-intravenous drug users.

The odds for HCV for incarcerated populations who had a history of shared drug injection or needle sharing was 7.57 times higher in comparison with those who had not a history of needle sharing. This association has also been reported in other studies, observing that the HCV infection is 2.58 and 4.06 times higher among those who had a history of needle sharing respectively in Brazil and Iran [21, 29].

Multiple strategies can be used to reduce the prevalence of HCV among incarcerated individuals. One approach, screen and treat, can be a cost-effective way to reduce ongoing HCV transmission before release into the community [30]. In a study conducted by He et al., universal opt-out HCV screening was a cost-effective way to reduce ongoing HCV disease and transmission benefiting both individuals and the community [30]. Screening has shown to be an effective first step in addressing HCV seropositivity and providing linkage to care but the practice has been under-utilized and inconsistent throughout incarcerated populations (Kronfli-International drug policy).

Another approach involves cross-collaboration among city, county and state public health departments to provide HCV positive incarcerated individuals proper linkage to care for treatment and education. This would involve making sure known IVDU (HCV positive and HCV negative) are released with the resources and education on where needle-exchange programs are located and how they can be utilized [31]. Needle exchange programs help reduce overall healthcare costs, reduce HCV transmission and reduce risky behaviours, such as needle sharing [32]. Currently, North Dakota has 4 needle exchange programs [33]. By increasing needle exchange programs throughout the state, more individuals (previous incarcerated and general population) can potentially benefit from these needle-exchange programs.

Currently all individuals admitted to the NDDOCR undergo screening, assessment and diagnosis for substance use disorders (SUD), HCV, and HVI. An individualized treatment plan is developed for SUD for each resident within 30 days of admission and treatment is offered to all individuals with chronic hepatitis C and an APRI of >0.5. The American Society of Addiction Medicine criteria are used to place each resident into the appropriate level of care for effective SUD treatment (https://www.asam.org/asam-criteria/about-the-asam-criteria). The NDDOCR utilizes evidence-based medication-assisted treatment for opioid use disorders and alcohol use disorders following the ACA-ASAM Joint Public Policy Correctional Policy on the Treatment of Opioid Use Disorders for Justice Involved Individuals (https://www. asam.org/docs/default-source/public-policy-statements/2018-joint-public-correctional-policy-on-the-treatment-of-opioid-use-disorders-for-justice-involved-individuals.pdf?sfvrsn=

26de41c2_2). NDDOCR utilizes the University of Cincinnati College of Education, Criminal Justice, and Human Services Cognitive-Behavioral Interventions for Substance Abuse (CBI-SA) curriculum in treatment of SUD. Potential opportunities for the state of North Dakota and NDDOCR to improvement of these disorders include fostering local needle exchanges and provide HCV treatment to all identified HCV patients.

Direct antiviral agents (DAAs) are an effective yet underutilized tool to help treat incarcerated individuals with HCV. DAA therapy has a high cost and coupled with prison turnover rates and transfers, makes proper administration difficult [34]. However, modelling studies have shown that DAA therapy administered to chronic HCV incarcerated individuals who are known IVDU and have a sentence greater than 16 weeks could decrease future HCV transmission by 26–70% depending on length of stay and biological uptake of the drug [35]. A modelling study performed a cost-effective analysis of DAAs in an Australian prison system and has shown that DAAs fell below the willingness to pay threshold for Australia making DAAs a value in terms of total healthcare expenditure [35]. While these studies are just models, utilizing DAAs in IVDU who are incarcerated can benefit the community upon their release.

This study has limitations that should be mentioned. The major drawback was no information about the participants' sexual histories, and there was a lack of behavioural information collected such as frequency or duration of injection drug use and their blood transfusion histories. Additionally, the crude categorization of alcohol consumption (yes/no) likely resulted in our observed results and an incomplete confounding adjustment. In this retrospective study, we had additional limitations related to the collection of some data. HCVRNA data is only available for a small fraction of our study population and was not incorporated into this study. Well-established risk factors such as receiving surgery, blood transfusion, family members with HCV infection were also not assessed in this population. Had this more detailed information been collected, we would have further adjusted our analyses, which may have changed our observed results.

## Conclusion

Our findings clearly indicated that the HCV prevalence in ND incarcerated populations is higher compared with the general population. Also, specific incarcerated populations have higher HCV infection, especially those who have a history of taking intravenous drugs and sharing needles or syringes. HCV screening and treatment and providing education programs to reduce high risk behaviours should be the priority and would be helpful tin reducing the HCV burden in incarcerated populations. Increased needle exchange programs throughout the state can help known IVDU post incarceration. These strategies can help reduce the HCV burden in incarcerated populations and have an indirect impact on reducing HCV prevalence in general populations. The results of this study clearly indicate that policy makers should consider more precautionary measures for preventing HCV in high-risk groups of populations during incarceration. Although it is not always possible to provide antiviral treatment to a large proportion of individuals during their incarceration periods because of their short stays, they still would benefit from HCV education and counselling programs if provided on a regular basis. In addition, the majority of HCV patients will suffered with liver problems, so it would be helpful to provide better treatment facilities for liver problems, which in turn would lead to delayed HCV mortality.

## Supporting information

**S1 Fig. Reported 1256 HCV cases by risk behavior.**
(TIFF)

**S1 Table. Sex and race/ethnicity groups by intervenes drug use in study population.**
(PDF)

## Acknowledgments

The authors thank the North Dakota Department of Corrections and Rehabilitation
(NDDOCR) and North Dakota Department of Health (NDDoH) for data management.

## Author Contributions

**Conceptualization:** Tracy K. Miller, John J. Hagan, Rick J. Jansen.

**Data curation:** Tracy K. Miller, Kodi Pinks, Grace Njau, John J. Hagan.

**Formal analysis:** Liton Chandra Deb, Rick J. Jansen.

**Funding acquisition:** Tracy K. Miller, Rick J. Jansen.

**Methodology:** Tracy K. Miller, Rick J. Jansen.

**Project administration:** John J. Hagan, Rick J. Jansen.

**Supervision:** Tracy K. Miller, Rick J. Jansen.

**Writing – original draft:** Liton Chandra Deb, Hannah Hove.

**Writing – review & editing:** Liton Chandra Deb, Hannah Hove, Tracy K. Miller, Kodi Pinks, Grace Njau, John J. Hagan, Rick J. Jansen.

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
