## [Decision Letter · Decision Letter 0]

24 Nov 2021

PONE-D-21-32680Epidemiology of Hepatitis C virus infection among incarcerated populations in North Dakota.PLOS ONE

Dear Dr. Jansen,

Thank you for submitting your manuscript to PLOS ONE. After careful consideration, we feel that it has merit but does not fully meet PLOS ONE’s publication criteria as it currently stands. Therefore, we invite you to submit a revised version of the manuscript that addresses the points raised during the review process. You may note that there is some conflicting advice from the two reviewers - one recommends updated international citations while the other suggests narrowing the focus to the US. I am inclined to recommend the latter, given the scope of the work described in the manuscript. While you should respond to all of the reviewer comments, it is reasonable to describe this alternate approach in justifying the removal of the international content.

We look forward to receiving your revised manuscript.

Kind regards,

Andrea Knittel

Academic Editor

PLOS ONE

Journal Requirements:

2. Please provide additional information regarding the considerations made for the prisoners included in this study. For instance, please discuss whether participants were able to opt out of the study and whether individuals who did not participate receive the same treatment offered to participants. In addition, if the need for consent was waived by the ethics committee, please include this information.

Reviewers' comments:

Reviewer's Responses to Questions

**Comments to the Author**

1. Is the manuscript technically sound, and do the data support the conclusions?

Reviewer #1: Partly

Reviewer #2: Yes

2. Has the statistical analysis been performed appropriately and rigorously? 

Reviewer #1: No

Reviewer #2: Yes

3. Have the authors made all data underlying the findings in their manuscript fully available?

Reviewer #1: Yes

Reviewer #2: Yes

4. Is the manuscript presented in an intelligible fashion and written in standard English?

Reviewer #1: Yes

Reviewer #2: Yes

5. Review Comments to the Author

Reviewer #1: Overall, this manuscript provides some useful data from a state correctional facility. The authors do a nice job of presenting 10 years of data which add to the current literature around HCV prevalence among incarcerated populations. However, there are numerous inconsistencies, grammatical errors and organizational challenges which need to be addressed.

Introduction:

Line 54: please insert comma after short illness; also this sentence needs to be referenced.

Lines 66-71: these sentences can be omitted since the following paragraph discussed HCV studies among correctional populations (and the authors could transition to this section by indicating that correctional populations have a disproportionate risk to HCV exposure and infection).

Line 75: suggest making this more specific to US populations (there have been several recent studies in other countries on HCV prevalence among correctional populations).

Line 79: please change to: derived from HCV surveillance data from eight states…

Line 81: 1.3 million of the 3.3 million populations? This doesn’t make sense, please clarify.

Lines 83-84: while the data presented certainly imply that correctional populations have much higher prevalence of HCV related risk factors (and studies have clearly documented this), the authors present very little data specific to risk factors. Please include studies that reference specific HCV risk factors in order to justify this.

Methods:

Line 91: what do the authors mean by “screening”? Does this mean testing? If so, what type of testing was performed?

Lines 92-93 (and elsewhere in the manuscript): please use more person-first language and avoid using pejorative terms such as “offenders”. Suggest using persons/individuals who are incarcerated.

Line 96: please change to: informed consent was not collected

Line 98: please change to: data used for this study were the…

Line 102: Again, please describe what screening means and what serological testing was performed.

Line 102: Was the behavioral data collected only through the study or is this information that the prison regularly collects from everyone?

Line 104: Do the authors mean length of total lifetime incarcerated?

Line 105: Please change to: age at first incarceration.

Line 107: How many correctional facilities were included? It would be helpful to provide a brief statement about the organizational structure of the NDDDOCR.

Line 115: Baby boomers are at higher risk, but not all are at high risk, please clarify.

Line 116: Hispanic is not a race so presumably the authors decided to combine race and ethnicity?

Line 117: Prevalence is not a rate, please change.

Line 119: Again, need to specify the testing performed.

Line 126: what do the authors mean by “biologically meaningful interactions”?

Line 126: Which two explanatory variables were thought to be potential confounders and why? Or do the authors mean that they assessed effects of confounders 2 explanatory variables at a time using the methods described? Please clarify.

Results:

Line 135: How were individuals diagnosed? Were the authors able to assess acute versus chronic infection? Was confirmatory testing prior to diagnosis done?

Line 136: Please do not begin a sentence with a number.

Line 137: Please change behaviors to behavioral. What behavioral information was missing (or were the data just incomplete)? Please be more explicit as to what data were missing and why these individuals were excluded.

Line 138: remove “in where” and insert a semi-colon between analysis and 1,1256.

Line 140: please use older age category rather than older people category. Same with younger people.

Line 141: Please change to the prevalence was (not is) much higher

Lines 142-143: this is very confusing as worded. Should be prevalence was higher in people with hx of IDU relative to those without such a hx. And being a new sentence for data related to sharing needles.

Table 1: did the authors collect specific information about frequency and duration of both alcohol and drug use? Especially for alcohol use, a dichotomous yes/no variable is not particular informative.

Line 148: please change to incarcerated individuals rather than peoples.

Lines 148-149: please insert commas for 1,256 and 6,027.

Table 2 with univariate analyses is not needed. The authors can just state what variables were selected, based on univariate analyses, for the multivariate model.

Line 156: change were to are.

Line 157: remove the parentheses around age. Same in the following sentence.

Line 159: add ‘s’ to male.

Line 160: change have to had.

Line 162: what does shared drug injection mean? How does this differ from needle sharing? Does the refer to sharing of drug injection equipment? How were these data collected?

Line 163: the finding related to alcohol may be due to the very imprecise manner in which this variable was measures (e.g. simple yes/no).

Line 166: add an ‘a’ in front of majority.

Discussion:

Line 180: since the authors collected data over 10 years, why didn’t they report annual HCV prevalence and conduct trend analyses to see how the prevalence changed over time?

Line 185: Should be HCV prevalence estimates.

Line 187: please change to: have been reported from other studies in countries such as…

Line 189: Being a new sentence: Overall, available data suggests…

Line 189: please change prisoners to individuals incarcerated in prison settings.

Line 192: this is the first reference to IDU. Previously, the authors have used the term intravenous drug use. Suggest using the acronym IDU consistently throughout.

Line 194: include citation for IDU statement.

Line 195: please be consistent with use of acronyms. Here the authors use IVDU for the first time.

Lines 195-197: please be careful with the use of the term rate. The authors are using prevalence estimates and prevalence is not a rate.

Line 204: please change peoples to individuals

Line 205: why is Birth capitalized?

Lines 205-208: this section should be integrated into the above section where the authors discuss their findings by age.

Lines 209-217: did the authors examine IDU history by gender? This may also partially explain the findings from this study. The authors should also note that this finding is somewhat contradictory to other studies that have documented higher HCV prevalence among men relative to women. Why didn’t the authors collect information on sexual history since they collected data on substance use history? This is a limitation.

Lines 218-223: As previously mentioned, the imprecise measurement of alcohol assumption likely contributed to this finding. The authors need to note this and explain why they reported such a crude measure for alcohol consumption. This is a limitation.

Lines 224-228: this is a weak discussion of the finding that HCV infection was higher among incarcerated Native Americans. As with gender, the authors could have examined history of IDU by race to see if more Native Americans reported IDU behavior.

Line 231: please change inmates to incarcerated individuals.

Line 233: change to non-injecting drugs

Lines 237-242: suggest integrating history of IDU and history of syringe sharing as these overlap. The authors also allude to shared drug injection and as previously noted, this is not clear.

Line 243: there needs to be some sort of transition sentence here. Also, the first part of this section is confusing. Do the authors mean to suggest providing sterile syringes while incarcerated? Their data do not suggest individuals are being infected with HCV prior to incarceration. Was IDU behavior during incarceration assessed? It would seem the authors mean to suggest that syringe access programs in the community, prior to individuals being incarcerated is (and indeed has been shown to be) an important prevention strategy. Most DOCs in the US would be extremely adverse to providing syringes to incarcerated individuals.

Also, this paragraph is unfocused and hard to follow. The authors present several potential interventions but there are significant variations re where and how these interventions are typically implemented. Syringe access is one. Treatment with DAAs is another, but entirely different since this is focused on treatment rather than prevention. And while the authors note that correctional facilities are “underutilized”, they make no mention of how the prohibitive costs of DAAs can be addressed. In fact, there is litigation around compelling correction systems to offer DAAs to their populations. However, most facilities resist offering treatment citing costs as the primary barrier. The authors need to engage in a more thoughtful discussion around all of this.

Line 253: here the authors mention incidence for the first time. Reducing incidence in correctional settings is very different than reduce prevalence (the latter of course depends on community responses while the former depends on preventing infection while incarcerated). Again, this paragraph is not well-organized. Test and treat is an important model worth its own section. Syringe access is separate from this and warrants another section. These are lumped together and neither are described sufficiently. Finally, the authors mention addiction treatment toward the end of this para and this is the first mention of treatment. Medication to treat opioid use disorder (MOUD) is an evidence based treatment approach which can reduce HCV incidence. But the authors make almost no mention of this. And, any discussion of MOUD should include a discussion of the fact that many correctional facilities do not offer MOUD for individuals with opioid use disorder.

Conclusion:

Line 274: Again, please do not refer to prevalence as a rate.

Line 276: the authors’ data documented higher HCV prevalence among females but this is not generalizable so the authors should not state that HCV is higher among incarcerated females vs males.

Line 277—see prior comments about being very clear about intervention strategies and which strategies should be implemented where and with which specific populations.

Line 287: the authors bring up liver problems but do not address this elsewhere in the paper. This seems like an afterthought rather than an important consideration for optimal care for incarcerated persons.

Reviewer #2: Jansen et al aimed to determine the prevalence of HCV infections among incarcerated people in a state prison system. Prison population is one of the most important targets to elaborate HCV micro-elimination pathways and these kinds of papers are fundamental for the scientific community.

However, there are some points to address on the work.

Methods

Regarding behavioral information (alcohol consumption, drug use, and needle sharing) did the authors establish a timeline (e.g. last 6 months etc) or just life history? (This could be a bias, given formers are different by active/recent users). Please, specify.

Results

L134. Please, put ‘table 1’ without brackets.

When discussing your results, please use ‘HCV antibody (Ab) positive’ or ‘HCV active infection’ instead of HCV positive or HCV infection. This would be less confusing for the reader.

Table 1 refers to plausible determinants of HCV in the Incarcerated population. At this point, it seems there is no reason do just report a description. I suggest authors to directly perform a comparative analysis.

Table 2 is reported as "univariate analysis of plausible determinants of HCV in Incarcerated population". It is not understandable if the authors are referring to antibody positivity or active infection. Please, specify. Furthermore, please report the p-values for your chi-squared test.

Discussion

Authors stated in Italy HCV prevalence is higher. However, the reference is old. Please, substitute the reference with Fiore et al. (doi: 10.1016/j.drugpo.2020.103055). Basing on this recent prospective study, HCV-Ab prevalence is 10.4%.

When coming to female population, the authors state ‘[…] higher prevalence of HCV infected women who

215 reported that their sexual partners were injection drug users’. I suggest adding some concepts: the female populations’ offenses maybe more frequently related to drugs and prostitution. Furthermore, these 2 offenses maybe related to each other (e.g. being sex worker to obtain drugs).

Moreover, I suggest authors to better comment NSP/OST usefulness (DOI: 10.1016/j.drugpo.2021.103407; DOI: 10.1002/14651858.CD012021), how educational programs may increase the cascade of care in prison population (https://doi.org/10.1016/j.drugpo.2018.04.003), the needing of a better linkage to care when coming to PWIDs (DOI: 10.1007/s10900-007-9083-3; DOI: 10.1016/j.idc.2018.02.001), and how DAAs changed the perspectives of HCV treatment among incarcerated patients (DOI: 10.1111/liv.14745; DOI: 10.1016/j.drugpo.2018.06.017)

Minor comments

There are a lot of typos inside the text (e.g. double brackets when referring to table 2, ‘peoples’, ‘p-value’ should be in lowercase italics, etc.). Please, carefully revise the text before resubmitting.

For advocacy reasons, do not use ‘inmates’ or ‘prisoners’. Please, substitute this term with ‘incarcerated people’, ‘incarcerated patients’, or ‘people who are incarcerated’.

In conclusion, the paper is worth to be shared with the scientific community, but still needs some adjustments before being ready to be published.

6. PLOS authors have the option to publish the peer review history of their article (what does this mean?). If published, this will include your full peer review and any attached files.

Reviewer #1: No

Reviewer #2: No

---

## [Author Response · Author response to Decision Letter 0]

9 Feb 2022

Review Comments to the Author

Reviewer #1: Overall, this manuscript provides some useful data from a state correctional facility. The authors do a nice job of presenting 10 years of data which add to the current literature around HCV prevalence among incarcerated populations. However, there are numerous inconsistencies, grammatical errors and organizational challenges which need to be addressed.

Introduction:

Line 54: please insert comma after short illness; also this sentence needs to be referenced.

comma and citation added

Lines 66-71: these sentences can be omitted since the following paragraph discussed HCV studies among correctional populations (and the authors could transition to this section by indicating that correctional populations have a disproportionate risk to HCV exposure and infection). 

omitted sentences and added transition sentence

Line 75: suggest making this more specific to US populations (there have been several recent studies in other countries on HCV prevalence among correctional populations). 

added “in the US” so line 77 to make it clear this was our focus.

Line 79: please change to: derived from HCV surveillance data from eight states… 

phrase modified as suggested.

Line 81: 1.3 million of the 3.3 million populations? This doesn’t make sense, please clarify. 

Modified to: “…was responsible for 1.3 million (39%) of the HCV burden within those states”

Lines 83-84: while the data presented certainly imply that correctional populations have much higher prevalence of HCV related risk factors (and studies have clearly documented this), the authors present very little data specific to risk factors. Please include studies that reference specific HCV risk factors in order to justify this.

Risk factors of HCV that are observed at a higher rate in incarcerated populations compared to the general population includes heavy alcohol drinking, IVDU, and needle sharing. Heavy alcohol drinking has been reported to increase the risk of HCV by 1.58 times [doi: 10.5812/hepatmon.31541], IVDU is reported in a meta-analysis study to increase the risk of HCV infection by 24 times [25] while needle sharing has been associated with an increased risk of 2.58 - 4.06 times [18, 27].

Methods:

Line 91: what do the authors mean by “screening”? Does this mean testing? If so, what type of testing was performed?

Yes, here it was measuring the antibody to HCV (anti-HCV) in a person’s serum. Added the phrase: “…negative when testing for antibodies to HCV (anti-HCV) in serum” to clarify.

Lines 92-93 (and elsewhere in the manuscript): please use more person-first language and avoid using pejorative terms such as “offenders”. Suggest using persons/individuals who are incarcerated.

Thanks, changed offenders to person or incarcerated person per suggestion.

Line 96: please change to: informed consent was not collected

Thanks. Changed.

Line 98: please change to: data used for this study were the…

Changed.

Line 102: Again, please describe what screening means and what serological testing was performed.

a. We perform reflex testing: 

i. Universal opt-out testing upon arrival for HCV antibody – less than two refusals per year!

ii. If antibody positive, viral load and genotype identification using the same sample.

Line 102: Was the behavioral data collected only through the study or is this information that the prison regularly collects from everyone?

This information was collected regularly by NDDOCR with the help of NDDoH. Modified second to last sentence to read: “All behavioural, medical and incarceration related information routinely were collected for each individual at the time of their entry into the correctional facilities.”

Line 104: Do the authors mean length of total lifetime incarcerated?

Yes, removed comma between lifetime and incarcerated.

Line 105: Please change to: age at first incarceration.

Changed as per suggestion.

Line 107: How many correctional facilities were included? It would be helpful to provide a brief statement about the organizational structure of the NDDDOCR.

Here only the North Dakota Correctional facilities were included.

Line 115: Baby boomers are at higher risk, but not all are at high risk, please clarify.

Changed sentence to read: “Age was categorized into three groups based on the previous literature detailing that persons born during 1945–1965 (i.e., baby boomers) and are at higher risk of HCV as a group because of potential contaminated blood exposures.”

Line 116: Hispanic is not a race so presumably the authors decided to combine race and ethnicity?

Yes, we changed all mentions of race to race/ethnicity.

Line 117: Prevalence is not a rate, please change.

Thanks for pointing out. Changed

Line 119: Again, need to specify the testing performed.

See line 102 response above.

Line 126: what do the authors mean by “biologically meaningful interactions”?

Changed the sentence to read: “Select interactions were also checked based on previously published observations and our study results.”

Line 126: Which two explanatory variables were thought to be potential confounders and why? Or do the authors mean that they assessed effects of confounders 2 explanatory variables at a time using the methods described? Please clarify.

The sentence was changed to clarify the method used: “Confounding effect of an explanatory variable was also evaluated by assigning the change of parameter estimates before and after removal of a variable from the model.”

Results:

Line 135: How were individuals diagnosed? Were the authors able to assess acute versus chronic infection? Was confirmatory testing prior to diagnosis done?

a. Diagnosis was based on incoming blood tests.

b. Acute vs chronic was determined by persistence of infection on 6 month recheck. Virtually 100% were chronic, as residents were admitted from county jail populations, where they had been in remand for more than 6 months pending trial/legal actions.

c. The reflex testing of positive antibodies for viral load and genotype served as the confirmatory tests. 

Line 136: Please do not begin a sentence with a number.

Thanks for pointing out this.

Line 137: Please change behaviors to behavioral. What behavioral information was missing (or were the data just incomplete)? Please be more explicit as to what data were missing and why these individuals were excluded.

Those individuals’ data were incomplete. They don’t have their drug use history and needle sharing information. Included the reasons for excluding

Line 138: remove “in where” and insert a semi-colon between analysis and 1,1256.

Thanks. Included.

Line 140: please use older age category rather than older people category. Same with younger people.

Thanks. Changed.

Line 141: Please change to the prevalence was (not is) much higher.

Thanks. Replaced.

Lines 142-143: this is very confusing as worded. Should be prevalence was higher in people with hx of IDU relative to those without such a hx. And being a new sentence for data related to sharing needles.

Thanks. Edited the sentence. It now reads: “Also, the prevalence was much higher in people who had a history of taking intravenous drugs (36.11%) compared with those without history of intravenous drugs (6.98%). Similarly, the prevalence was much higher for people with sharing needles (46.82%) compared with those that did not have any history of needle sharing (10.41%) ((Table 2).”

Table 1: did the authors collect specific information about frequency and duration of both alcohol and drug use? Especially for alcohol use, a dichotomous yes/no variable is not particular informative.

This study used the secondary data that’s why we have a limitation that we don’t have chance to collect those frequency and duration information.

Line 148: please change to incarcerated individuals rather than peoples.

Thanks. Changed

Lines 148-149: please insert commas for 1,256 and 6,027.

Inserted.

Table 2 with univariate analyses is not needed. The authors can just state what variables were selected, based on univariate analyses, for the multivariate model.

We have combined table 1 and 2 to eliminate overlap of information.

Line 156: change were to are.

changed

Line 157: remove the parentheses around age. Same in the following sentence.

removed

Line 159: add ‘s’ to male.

added

Line 160: change have to had.

changed

Line 162: what does shared drug injection mean? How does this differ from needle sharing? Does the refer to sharing of drug injection equipment? How were these data collected?

a. This data was collected from resident during intake medical interview. The question is “Now or in the past, have you shared injection drugs or needles with anyone else? This includes needles, syringes, spoons, filters or rinse water?” The intention is to identify opportunities to transmit HCV during I the injection process. 

Line 163: the finding related to alcohol may be due to the very imprecise manner in which this variable was measures (e.g. simple yes/no).

Thanks for pointing out. Possibly could the reason and we have mentioned this in the limitations in the discussion: “Additionally, the crude categorization of alcohol consumption (yes/no) likely resulted in our observed results and an incomplete confounding adjustment”

Line 166: add an ‘a’ in front of majority.

included

Discussion:

Line 180: since the authors collected data over 10 years, why didn’t they report annual HCV prevalence and conduct trend analyses to see how the prevalence changed over time?

We have now included annual HCV prevalence for 2009 – 2018 and included a figure with a trend line and logistic regression p-value (0.0542).

Line 185: Should be HCV prevalence estimates.

Thanks. Changed.

Line 187: please change to: have been reported from other studies in countries such as…

Changed.

Line 189: Being a new sentence: Overall, available data suggests…

Thanks. Included a new sentence.

Line 189: please change prisoners to individuals incarcerated in prison settings.

Included.

Line 192: this is the first reference to IDU. Previously, the authors have used the term intravenous drug use. Suggest using the acronym IDU consistently throughout.

Thanks. Added.

Line 194: include citation for IDU statement.

Included.

Line 195: please be consistent with use of acronyms. Here the authors use IVDU for the first time.

Thanks edited.

Lines 195-197: please be careful with the use of the term rate. The authors are using prevalence estimates and prevalence is not a rate.

Thanks. Omitted rate.

Line 204: please change peoples to individuals

Replaced

Line 205: why is Birth capitalized?

Thanks. Changed.

Lines 205-208: this section should be integrated into the above section where the authors discuss their findings by age.

Thanks. Integrated with previous section.

Lines 209-217: did the authors examine IDU history by gender? This may also partially explain the findings from this study. The authors should also note that this finding is somewhat contradictory to other studies that have documented higher HCV prevalence among men relative to women. Why didn’t the authors collect information on sexual history since they collected data on substance use history? This is a limitation.

We have added an additional table looking at IDU by gender and race/ethnicity as supporting information. This study is based on secondary data, so there was no way to collect sexual history. That’s true it’s a limitation for this study and has been listed as such at the end of the discussion.

Lines 218-223: As previously mentioned, the imprecise measurement of alcohol assumption likely contributed to this finding. The authors need to note this and explain why they reported such a crude measure for alcohol consumption. This is a limitation.

We have now mentioned this in the limitations in the discussion: “Additionally, the crude categorization of alcohol consumption (yes/no) likely resulted in our observed results and an incomplete confounding adjustment”

Lines 224-228: this is a weak discussion of the finding that HCV infection was higher among incarcerated Native Americans. As with gender, the authors could have examined history of IDU by race to see if more Native Americans reported IDU behavior.

We have added an additional supporting information table looking at IDU by gender and race/ethnicity.

Line 231: please change inmates to incarcerated individuals.

Changed.

Line 233: change to non-injecting drugs

Changed 

Lines 237-242: suggest integrating history of IDU and history of syringe sharing as these overlap. The authors also allude to shared drug injection and as previously noted, this is not clear.

a. IDU and sharing of syringes were data collected with different questions and the multivariable regression results indicated that both are independent risk factors for testing positive.

Line 243: there needs to be some sort of transition sentence here. Also, the first part of this section is confusing. Do the authors mean to suggest providing sterile syringes while incarcerated? Their data do not suggest individuals are being infected with HCV prior to incarceration. Was IDU behavior during incarceration assessed? It would seem the authors mean to suggest that syringe access programs in the community, prior to individuals being incarcerated is (and indeed has been shown to be) an important prevention strategy. Most DOCs in the US would be extremely adverse to providing syringes to incarcerated individuals.

This paragraph was edited.

Also, this paragraph is unfocused and hard to follow. The authors present several potential interventions but there are significant variations re where and how these interventions are typically implemented. Syringe access is one. Treatment with DAAs is another, but entirely different since this is focused on treatment rather than prevention. And while the authors note that correctional facilities are “underutilized”, they make no mention of how the prohibitive costs of DAAs can be addressed. In fact, there is litigation around compelling correction systems to offer DAAs to their populations. However, most facilities resist offering treatment citing costs as the primary barrier. The authors need to engage in a more thoughtful discussion around all of this.

This section has been edited. Interventions were placed into separate paragraphs, recent references were used and cost barriers addressed for DAAs. 

Line 253: here the authors mention incidence for the first time. Reducing incidence in correctional settings is very different than reduce prevalence (the latter of course depends on community responses while the former depends on preventing infection while incarcerated). Again, this paragraph is not well-organized. Test and treat is an important model worth its own section. Syringe access is separate from this and warrants another section. These are lumped together and neither are described sufficiently. Finally, the authors mention addiction treatment toward the end of this para and this is the first mention of treatment. Medication to treat opioid use disorder (MOUD) is an evidence based treatment approach which can reduce HCV incidence. But the authors make almost no mention of this. And, any discussion of MOUD should include a discussion of the fact that many correctional facilities do not offer MOUD for individuals with opioid use disorder.

This section was rephrased and paragraphs were separated.

Conclusion:

Line 274: Again, please do not refer to prevalence as a rate.

Thanks. Removed

Line 276: the authors’ data documented higher HCV prevalence among females but this is not generalizable so the authors should not state that HCV is higher among incarcerated females vs males.

Thanks. Removed that part.

Line 277—see prior comments about being very clear about intervention strategies and which strategies should be implemented where and with which specific populations.

We have edited these paragraphs and added additional citations as supporting evidence.

Line 287: the authors bring up liver problems but do not address this elsewhere in the paper. This seems like an afterthought rather than an important consideration for optimal care for incarcerated persons.

We have mentioned HCV as a major cause of liver disease in the introduction and mention it in the conclusion as a way to identify significant impacts HCV prevention or treatment may have on outcomes in this population.

Reviewer #2: Jansen et al aimed to determine the prevalence of HCV infections among incarcerated people in a state prison system. Prison population is one of the most important targets to elaborate HCV micro-elimination pathways and these kinds of papers are fundamental for the scientific community.

However, there are some points to address on the work.

Methods

Regarding behavioral information (alcohol consumption, drug use, and needle sharing) did the authors establish a timeline (e.g. last 6 months etc) or just life history? (This could be a bias, given formers are different by active/recent users). Please, specify. 

This information routinely was collected at incarceration by North Dakota Department of Corrections and Rehabilitation (NDDOCR) and was a history of exposure with no timeline specified. That’s true that could be a bias due to active or recent users but there is no way get those information by specific time period using our data. We have mentioned it in our limitation. Thanks for pointing out this.

Results

L134. Please, put ‘table 1’ without brackets. Brackets omitted

When discussing your results, please use ‘HCV antibody (Ab) positive’ or ‘HCV active infection’ instead of HCV positive or HCV infection. This would be less confusing for the reader. Thanks for the correction.

Table 1 refers to plausible determinants of HCV in the Incarcerated population. At this point, it seems there is no reason do just report a description. I suggest authors to directly perform a comparative analysis. 

We have combined table 1 and 2 to eliminate overlap of information.

Table 2 is reported as "univariate analysis of plausible determinants of HCV in Incarcerated population". It is not understandable if the authors are referring to antibody positivity or active infection. Please, specify. Thanks for mentioning that. Changed to HCV antibody (Ab).

 Furthermore, please report the p-values for your chi-squared test.

Thanks. P-value included in the table.

Discussion

Authors stated in Italy HCV prevalence is higher. However, the reference is old. Please, substitute the reference with Fiore et al. (doi: 10.1016/j.drugpo.2020.103055). Basing on this recent prospective study, HCV-Ab prevalence is 10.4%. Omitted the previous one and added the new citation.

When coming to female population, the authors state ‘[…] higher prevalence of HCV infected women who

215 reported that their sexual partners were injection drug users’. I suggest adding some concepts: the female populations’ offenses maybe more frequently related to drugs and prostitution. Furthermore, these 2 offenses maybe related to each other (e.g. being sex worker to obtain drugs). Thanks for pointing out this important note. Added.

Moreover, I suggest authors to better comment NSP/OST usefulness (DOI: 10.1016/j.drugpo.2021.103407; DOI: 10.1002/14651858.CD012021), how educational programs may increase the cascade of care in prison population (https://doi.org/10.1016/j.drugpo.2018.04.003), the needing of a better linkage to care when coming to PWIDs (DOI: 10.1007/s10900-007-9083-3; DOI: 10.1016/j.idc.2018.02.001), and how DAAs changed the perspectives of HCV treatment among incarcerated patients (DOI: 10.1111/liv.14745; DOI: 10.1016/j.drugpo.2018.06.017)

Minor comments

There are a lot of typos inside the text (e.g. double brackets when referring to table 2, ‘peoples’, ‘p-value’ should be in lowercase italics, etc.). Please, carefully revise the text before resubmitting.

For advocacy reasons, do not use ‘inmates’ or ‘prisoners’. Please, substitute this term with ‘incarcerated people’, ‘incarcerated patients’, or ‘people who are incarcerated’.

In conclusion, the paper is worth to be shared with the scientific community, but still needs some adjustments before being ready to be published.

Thank you for your thoughtful comments and references. These comments have been addressed and additional references have been utilized. Substitutions have been made in regard to ‘inmates’ and ‘prisoners’ and typos adjusted.

---

## [Decision Letter · Decision Letter 1]

7 Mar 2022

PONE-D-21-32680R1Epidemiology of Hepatitis C virus infection among incarcerated populations in North Dakota.PLOS ONE

Dear Dr. Jansen,

Thank you for submitting your manuscript to PLOS ONE. After careful consideration, we feel that it has merit but does not fully meet PLOS ONE’s publication criteria as it currently stands. Therefore, we invite you to submit a revised version of the manuscript that addresses the points raised during the review process.

We look forward to receiving your revised manuscript.

Kind regards,

Jee-Fu Huang, M.D., Ph.D.

Academic Editor

PLOS ONE

Additional Editor Comments:

The study aimed to clarify HCV infections and risk factors in incarcerated people in the US by using the large-scale database. However, previous studies have been vigorously investigating the issue. The cross-sectional features of the study and the lack of HCVRNA, both quantitative and genotype approaches, much limited the results and interpretations of the study. The results and the conclusions did not appear to address more novelty in the current knowledge. Therefore,

1. The Authors are encouraged to provide HCVRNA data for the epidemiological features and also put the RNA data into risk analysis.

2. The other well-established risk factors such as receiving surgery, blood transfusion, family members of HCV infection, etc. could be addressed.

3. The proposed strategies for risk reduction in the special population could be discussed in the discussion section.

---

## [Author Response · Author response to Decision Letter 1]

8 Mar 2022

1. The Authors are encouraged to provide HCVRNA data for the epidemiological features and also put the RNA data into risk analysis.

1. This is a retrospective study and so RNA presence data is only available for patients in the care of NDDOCR on or after 09/2015, but not prior to that. In addition, the original IRB did not specify obtaining this variable and so an additional IRB protocol would need to be submitted and approved to have access to this data. This would extend our resubmission beyond the allotted journal timeframe.

2. The other well-established risk factors such as receiving surgery, blood transfusion, family members of HCV infection, etc. could be addressed.

2. Surgery, blood transfusion, family member info was not collected and so is not available for this population.

3. The proposed strategies for risk reduction in the special population could be discussed in the discussion section.

3. We have discuss strategies that have been implemented in justice involved populations in the discussion pages 14-15. We have also added the following paragraph after line 283 in the discussion section as well:

Currently all individuals admitted to the NDDOCR undergo screening, assessment and diagnosis for substance use disorders (SUD), HCV, and HVI. An individualized treatment plan is developed for SUD for each resident within 30 days of admission and treatment is offered to all individuals with chronic hepatitis C and an APRI of >0.5. The American Society of Addiction Medicine criteria are used to place each resident into the appropriate level of care for effective SUD treatment (https://www.asam.org/asam-criteria/about-the-asam-criteria). The NDDOCR utilizes evidence-based medication-assisted treatment for opioid use disorders and alcohol use disorders following the ACA-ASAM Joint Public Policy Correctional Policy on the Treatment of Opioid Use Disorders for Justice Involved Individuals (https://www.asam.org/docs/default-source/public-policy-statements/2018-joint-public-correctional-policy-on-the-treatment-of-opioid-use-disorders-for-justice-involved-individuals.pdf?sfvrsn=26de41c2_2). NDDOCR utilizes the University of Cincinnati College of Education, Criminal Justice, and Human Services Cognitive-Behavioral Interventions for Substance Abuse (CBI-SA) curriculum in treatment of SUD. Potential opportunities for the state of North Dakota and NDDOCR to improvement of these disorders include fostering local needle exchanges and provide HCV treatment to all identified HCV patients.

---

## [Editor Report · Decision Letter 2]

10 Mar 2022

PONE-D-21-32680R2Epidemiology of Hepatitis C virus infection among incarcerated populations in North Dakota.PLOS ONE

Dear Dr. Jansen,

Thank you for submitting your manuscript to PLOS ONE. We invite you to submit a revised version of the manuscript that addresses the points raised during the review process after your responses to the previous comments.

We look forward to receiving your revised manuscript.

Kind regards,

Jee-Fu Huang, M.D., Ph.D.

Academic Editor

PLOS ONE

Journal Requirements:

Additional Editor Comments:

The Authors should put the lack of HCVRNA data and the commonly-observed risk factors into the study limitations.
---

## [Author Response · Author response to Decision Letter 2]

10 Mar 2022

1. The Authors should put the lack of HCVRNA data and the commonly-observed risk factors into the study limitations.

Answer: We have added the following to the limitations paragraph to the end of the discussion:

In this retrospective study, we had additional limitations related to the collection of some data. HCVRNA data is only available for a small fraction of our study population and was not incorporated into this study. Well-established risk factors such as receiving surgery, blood transfusion, family members with HCV infection were also not assessed in this population.

---

## [Editor Report · Decision Letter 3]

14 Mar 2022

Epidemiology of Hepatitis C virus infection among incarcerated populations in North Dakota.

PONE-D-21-32680R3

Dear Dr. Jansen,

We’re pleased to inform you that your manuscript has been judged scientifically suitable for publication and will be formally accepted for publication once it meets all outstanding technical requirements.

Kind regards,

Jee-Fu Huang, M.D., Ph.D.

Academic Editor

PLOS ONE
---

## [Editor Report · Acceptance letter]

21 Mar 2022

PONE-D-21-32680R3 

Epidemiology of Hepatitis C virus infection among incarcerated populations in North Dakota. 

Dear Dr. Jansen:

I'm pleased to inform you that your manuscript has been deemed suitable for publication in PLOS ONE. Congratulations! Your manuscript is now with our production department. 

Kind regards, 

on behalf of

Dr. Jee-Fu Huang 

Academic Editor

PLOS ONE